# Existing Adversarial Large Language Model Unlearning Evaluations Are Inconclusive

## Abstract

Unlearning seeks to remove sensitive knowledge from large language models, with success often judged through adversarial evaluations. In this work, we critically examine these evaluation practices and reveal key limitations that undermine their reliability. First, we show that adversarial evaluations introduce new information into the model, potentially masking true unlearning performance by re-teaching the model during evaluation. Second, we show that evaluation outcomes vary significantly across tasks, undermining the generalizability of current evaluation methods. Collectively, these issues suggest that existing evaluations risk mischaracterizing unlearning success (or failure). To address this, based on our empirical findings, we propose two principles—*minimal information injection* and *downstream task awareness*—for future evaluations.

## 1 Introduction

Despite the impressive capabilities of large language models (LLMs), their widespread use introduces significant challenges related to responsible use, particularly regarding their retention of sensitive knowledge (Carlini et al., 2024). To address these risks, the field of *unlearning* aims to propose methods that selectively remove knowledge from LLMs, providing a means to excise sensitive content without the costs of retraining models from scratch (Li et al., 2024b; Zhang et al., 2024).

Owing to the high-stakes nature of deploying LLMs in safety-critical contexts, it is crucial to design robust evaluations that reliably determine whether unlearning has been successful. Much existing work focuses on *adversarial evaluation methods* that probe models for knowledge of unlearned content (Łucki et al., 2024; Schwarzschild et al., 2024). While common in practice, we find that three existing adversarial evaluations—finetuning attacks (which manipulate model parameters), input-space attacks (which extract information via prompting), and memorization detectors (which are based on information compression)—suffer from two critical shortcomings. First, they often inject additional information into the model, making it difficult to decouple preexisting knowledge from artifacts of the evaluation process. Second, they frequently employ task-dependent metrics (e.g., multiple-choice accuracy) that fail to account for the diversity of model use cases.

We argue that these limitations undermine the validity of current evaluations, particularly in their capacity to inform model deployment in real-world, safety-critical applications. To this end, our analysis informs the proposal of two principles to guide future unlearning evaluations. First, the *minimal information injection* principle stipulates that evaluations should minimize the amount of information injected via prompts or weight editing. Second, the *downstream task awareness* principle states that future evaluations should anticipate a wider scope of model use, including open-ended generation. In proposing these principles and in demonstrating how they can be applied in practice, we aim to lay the groundwork for more reliable and actionable assessments of unlearning effectiveness.

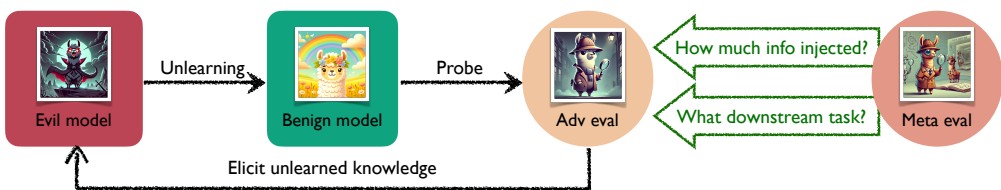

Figure 1: **The evaluation cycle of unlearned models.** Adversarial unlearning evaluations seek to determine whether an unlearned model retains sensitive data after unlearning. In this paper, we show that existing evaluations offer inconclusive results and we propose two principles to improve reliability and ensure future evaluations more accurately reflect true forgetting.

## 2 Related work

Prior work defines unlearning in several distinct ways (Xu et al., 2024; Thudi et al., 2022). Exact unlearning aims to produce a model that is equivalent to a copy of that model trained only on a retain set, which excludes targeted information (Yan et al., 2022). Approximate unlearning instead aims to remove the statistical influence of targeted data, using a definition similar to differential privacy (Guo et al., 2019; Graves et al., 2021). Heuristic unlearning approaches the problem more abstractly, training models— similar to model alignment (Ouyang et al., 2022)—to refuse to generate targeted content. Each of these paradigms comes equipped with their own evaluation protocols, complicating the task of comparing unlearning algorithms (Thaker et al., 2024; Scholten et al., 2024).

**Exact and approximate unlearning.** Several recent pieces of legislation motivate the design of unlearning algorithms. Regulations such as the EU's GDPR (Article 17) (Union, 2016), the UK's GDPR (UK Government, 2021), California's CCPA/CPRA (OAG, 2021), and Canada's proposed CPPA (Government of Canada, 2022) grant individuals the "right to be forgotten." To this end, exact unlearning aims to obtain a model identical to a model trained without a particular piece of data (Ullah et al., 2021; Bourtoule et al., 2021; Golatkar et al., 2020), whereas approximate unlearning aims to reduce data influence without guaranteeing complete removal (Ginart et al., 2019; Dwork et al., 2014; Izzo et al., 2021; Sekhari et al., 2021). In the context of LLMs, learning to "forget" concepts has resulted in several finetuning approaches to approximate unlearning (Eldan & Russinovich, 2023; Liu et al., 2025). Accompanying these algorithms are various datasets curated to benchmark unlearning in vision and language tasks (Ma et al., 2024; Shi et al., 2024; Jin et al., 2024). One notable corpus that we use throughout this work is the Task of Fictitious Unlearning (TOFU) dataset, which contains question-answer pairs about distinct, fictitious authors (Maini et al., 2024).

**Heuristic unlearning.** Heuristic unlearning aims to train models via preference optimization or representation-based finetuning to refuse to generate harmful information (Fan et al., 2024; Zhang et al., 2024; Lynch et al., 2024). Tamper-resistant methods, which train models to resist finetuning attacks (Che et al., 2024), have also shown promise when applied to unlearning problem settings (Tarun et al., 2023; Tamirisa et al., 2024). Evaluations of heuristic unlearning often rely on safety-oriented datasets, including variants of the Weapons of Mass Destruction Proxy (WMDP) benchmark (Li et al., 2024b), which focus on biological, chemical, cybersecurity risks.

**Evaluating unlearning.** A central component of unlearning is to determine whether a model has truly forgotten targeted information. To do so, several papers propose adversarial elicitation attacks (Łucki et al., 2024; Lynch et al., 2024; Deeb & Roger, 2024; Li et al., 2024a). Che et al. (2024) suggest that finetuning attacks tend to upper bound the success rate of adversarial prompting as well as non-adversarial detection methods, including latent-space probing. Relatedly, Schwarzschild et al. (2024) propose the adversarial compression ratio as a metric for LLM memorization. This work, and others in this spirit (Nasr et al., 2023; Ippolito et al., 2023), share a common aim: to demonstrate that the forget set is no longer memorized and that the retain set remains learned.

Several recent studies examine the fragility of LLM unlearning evaluations. Hu et al. (2025) find that an unlearned model can be retrained on a small, unrelated dataset to output harmful knowledge that it had supposedly forgotten. Thaker et al. (2024) also show that unlearning algorithms tend to overfit to narrowly

defined retain and forget sets, and that non-adversarial query changes tend to significantly change unlearning success metrics. In contrast, we focus our analysis on stronger finetuning and adversarial attacks, both of which are pervasive in the unlearning literature (Liu et al., 2025).

## 3 Preliminaries

**Terminology.** We use the following standard terminology to describe unlearning datasets and models. We refer to the dataset intended for removal as the *forget set*, and to the remainder of the training data, which the model should retain, as the *retain set*. The model before unlearning is the *base model*, the result of unlearning is the *unlearned model*, and a model trained from scratch on only the retain set is the *retain model*. In finetuning-based attacks, we call the model obtained by finetuning the unlearned model the *relearned model*, and the corresponding procedure is called *relearning*. In this context, a *spurious correlation* refers to a statistical dependency between the retain and forget sets that arises from incidental artifacts rather than the intended semantics of the data, enabling the model to generalize from retain examples to forget examples without truly learning the underlying task.

### 3.1 Unlearning evaluation methods

Drawing from the taxonomy laid out by Che et al. (2024), we center our analysis on prominent methods from three broad, representative classes of unlearning evaluations: finetuning attacks, input-space attacks, and memorization detectors. In the following subsections, we describe each evaluation, justify its choice, and provide in-depth preliminaries and notation specific to each attack.

**Finetuning attacks.** Finetuning attacks allow an adversary to finetune the unlearned model on several specifically chosen samples, a process we refer to as relearning. In practice, these training samples can be drawn from either the retain set or the forget set; in this work, we focus on samples drawn from the retain set, as they give stronger evidence of unlearning success or failure. To support this choice, consider the work of Łucki et al. (2024), who find that an unlearned model trained via RMU (Li et al., 2024b) to forget WDMP-Bio achieves only 29.9% accuracy on questions from this forget set. However, after relearning on just *five* samples from the retain set, the accuracy on the forget set spikes to 62.4%, whereas the base model only achieves 64.4% accuracy before unlearning anyway. This indicates that finetuning on a remarkably small subset of the retain set is sufficient to recover nearly all the accuracy of the base model, which evinces relatively weak unlearning in this case. We note that finetuning attacks establish an upper bound on the success rate of other adversarial evaluations, as they offer the greatest flexibility by directly modifying the model's weights (Che et al., 2024).

**Input-space attacks.** Whereas finetuning attacks edit model internals to probe relearning capabilities, input-space attacks seek to elicit supposedly unlearned text via prompting. In this paper, we consider the Enhanced GCG attack, which is a representative algorithm in this category (Łucki et al., 2024). The objective of Enhanced GCG is to optimize a single adversarial prompt, which, when prepended to any prompt in the forget set, facilitates the elicitation of unlearned knowledge. To operationalize this attack, let $M$ and $M_U$ denote a base and unlearned model, respectively, and let $D$ denote a given dataset, which can be chosen somewhat arbitrarily, but is often taken to be a subset of the retain set or the forget set. Enhanced GCG then seeks to solve the following problem:

$$\underset{x:|x|\leq k}{\text{maximize}} \quad \frac{1}{|D|} \sum_{y \in D} \log \Pr(x||y; M_U) + \lambda(x; D, M, M_U). \tag{1}$$

Here, $\cdot||\cdot$ denotes the concatenation operator, $x$ is the optimization variable, and $k$ is typically chosen to be 100. $\Pr(x||y; M_U)$ denotes the probability of the unlearned model generating a piece of knowledge that depends on $y$, and $\lambda(x; D, M, M_U)$ term is a regularization term that encourages the prefix $x$ be such that the internal representations of $M_U$ are relatively close to those of $M$ (Thompson & Sklar, 2024). Given a solution $x^\star$ to equation 1, one can evaluate $M_U$ on a held-out set.

**Memorization detectors.** Memorization detection is a fundamental piece of any unlearning evaluation pipeline. While various metrics exist for quantifying memorization, one prominent metric is the adversar-

ial compression ratio (ACR) (Schwarzschild et al., 2024). While not typically framed as an unlearning metric, the ACR is used to evince failures to forget information across various unlearned models. For example, Schwarzschild et al. (2024) highlight that the ACR of information that Eldan & Russinovich (2023) try to remove from an LLM remains unchanged. Given a model $M$ (either a base or unlearned model), the ACR of a string $y$ is defined as

$$\text{ACR}(M, y) = |y|/|x^\star|, \quad \text{where } x^* \in \arg\min |x| \text{ s.t } M(x) = y. \tag{2}$$

Here $|\cdot|$ denotes the length of a sequence of tokens, and $M(x) = y$ indicates that $M$ generates $y$ in response to a prompt $x$ under greedy decoding. To implement a complete evaluation of unlearning via the ACR, we compare the ACR measured separately for both the retain and forget sets to measure unlearning success on a per-sample basis.

### 3.2 Datasets, architectures, and unlearning algorithms

We use a variety of standard unlearning algorithms, datasets, and LLM architectures to facilitate our analysis of the effectiveness of current unlearning evaluations. Following Łucki et al. (2024), we use Zephyr-7B (Tunstall et al., 2023) as the base model in many of our experiments, and to offer points of comparison, we also evaluate Phi-1.5 (Li et al., 2023) and Llama-3.2-1B-Instruct models (Dubey et al., 2024) when applicable. All chat models are evaluated with empty system prompts. To obtain unlearned models, we use two standard algorithms: representation misdirection for unlearning (RMU) (Li et al., 2024b) and negative preference optimization (NPO) (Zhang et al., 2024). When evaluating unlearning, we use the standard splits of both TOFU (Maini et al., 2024) and WMDP (Li et al., 2024b), unless otherwise stated. Our evaluations of finetuning attacks also use a newly curated multiple-choice question (MCQ) version of TOFU, which we call TOFU-MCQ. This data comprises 2000 generated MCQs—ten MCQs for each of TOFU's 200 ficitious authors. We evaluate these behaviors on Phi-1.5, which, crucially, was released *before* the curation of the TOFU.

## 4 Pitfalls of adversarial unlearning evaluations

We begin by identifying and analyzing the shortcomings of the first two adversarial evaluation methods discussed in §3.1: finetuning attacks and input-space attacks. Our analysis centers on two key questions, which motivate the recommendations we make in later sections below.

*Question 1*: Does the information elicited by an adversarial evaluation reflect a failure to unlearn the forget set or new information internalization introduced by the attack process itself?

*Question 2*: To what extent is the effectiveness of adversarial evaluations sensitive to the formatting of the task (e.g., MCQs versus open-ended generation)?

The first question posits two possible sources for the information elicited in an unlearning evaluation: text that the unlearned model failed to forget, and text introduced inadvertently during evaluation. While identification of the first source conclusively points to ineffective unlearning, distinguishing it from the second is essential for correctly attributing failure and avoiding false positives in evaluation. The second question, on the other hand, concerns the sensitivity of standard evaluations. If evaluating unlearning via open-ended generation yields significantly different conclusions than an equivalent evaluation performed with MCQs, it suggests that conclusions about unlearning success may be tightly coupled to the chosen task format, limiting their reliability. We consider these questions for finetuning attacks in §4.1 and for input-space attacks in §4.2.

### 4.1 Finetuning attacks

Finetuning-based unlearning evaluations seek to adjust an unlearned model by training on a minimal number of samples. The resulting relearned model is then evaluated to determine whether it contains knowledge about the forget set. The fidelity of this evaluation rests on two key assumptions: (1) the relearned model should not generalize to the forget data purely from training on the retain data, and (2) the evaluator knows the data format used by the unlearning algorithm. In the remainder of this subsection, we empirically show that neither of these assumptions may hold in practice.

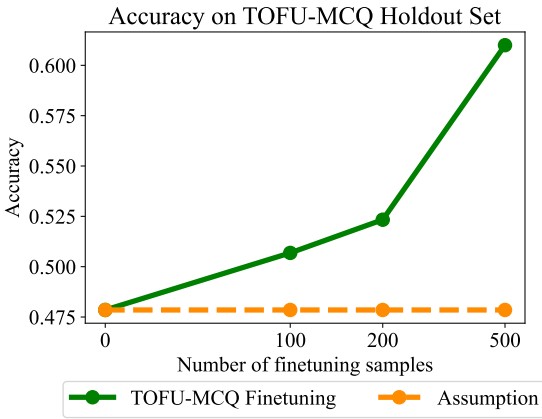

Figure 2: **Evidence of spurious generalization.** Finetuning attacks significantly improve accuracy on TOFU's test set. This finding indicates that it may be possible to spuriously generalize between the retain and forget sets in finetuning-based unlearning evaluations.

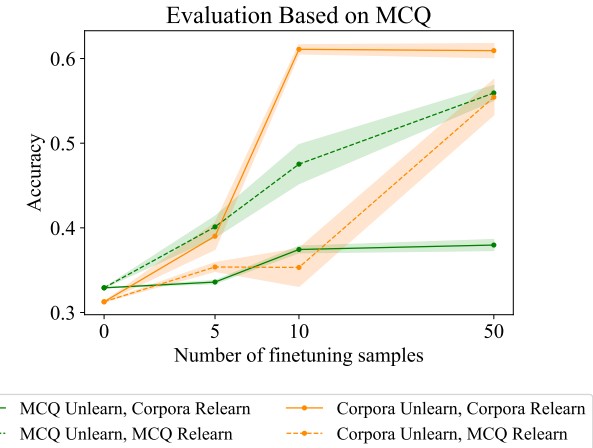

Figure 3: **Evidence of information injection.** Relearning effectiveness varies significantly when the data format of the downstream task generation format differs between evaluation and deployment.

**Evidence of spurious generalization.** Unlearning evaluations implicitly assume that the retain and forget sets are disjoint and independent. If this property does not hold, successful unlearning may be impossible, since unlearning the forget set may diminish a model's knowledge of the retain set, and conversely, finetuning on the retain set may result in relearning the forget set. To test whether this form of generalization manifests in existing finetuning evaluations, we design the following experiment. We first finetune Phi-1.5, which was released *before* the curation of the TOFU, on subsets of TOFU-MCQ containing 100, 200, and 500 instances. Next, we evaluate these finetuned models on held-out data from TOFU-MCQ. Since each instance in TOFU-MCQ comprises information about distinct people, one would expect generalization between the finetuning and held-out sets to be impossible. However, as shown in Fig. 2, we find that finetuning significantly improves accuracy on the held-out set. Specifically, by finetuning on several hundred samples, accuracy on the held-out set increases by up to 18%. This implies that TOFU, a widely used unlearning benchmark, contains spurious correlations that facilitate generalization between the retain and forget sets. In other words, finetuning a model that has never seen the forget set on retain data only may introduce knowledge about the forget set into the model, complicating the task of identifying successful unlearning.

**Evidence of data formatting dependence.** A critical, yet often overlooked aspect of unlearning is the role played by prior knowledge. Both unlearning and evaluation require a (somewhat arbitrary) choice of the unlearning algorithm (e.g., RMU (Li et al., 2024b) or DPO (Zhang et al., 2024)), the data format (e.g., MCQ or full corpora), and the evaluation metric. To probe the relationship between evaluation results and data format, we consider the following experiment. We first take two models: one is unlearned via NPO on WMDP-Bio MCQ data and the other is unlearned via RMU on WMDP-Bio full corpora data. Next, we finetune both models separately on the retain set of these data subsets, yielding four distinct relearned checkpoints. Figure 3 shows that relearning effectiveness varies significantly when the finetuning data differs in format or structure from the original unlearning data, even if they encode the same information. In particular, we observe that relearning tends to require fewer samples when the unlearning and relearning data formats match. This implies that successful evaluations may require prior knowledge about the original unlearning algorithm.

To better illustrate the impact of prior knowledge, consider the following hypothetical example (Figure 4) which intuitively demonstrates that the relearning path may depend on the unlearning algorithm and data format with a hypothetical example (Figure 4). Specifically, this example illustrates that numerous relearned finetuning trajectories are possible, and that prior knowledge of the forget set may influence the extent to which a relearned model approaches a checkpoint similar to the base model. We note that many studies

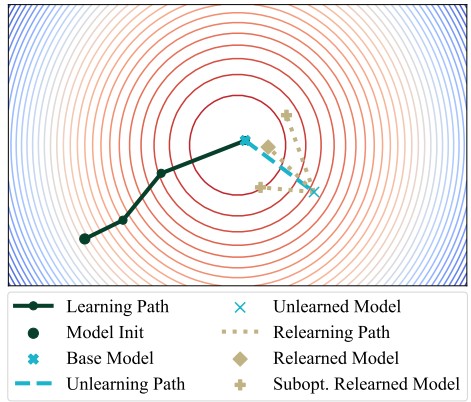

Figure 4: **Relearning trajectories.** A hypothetical case in which two phenomena occur: (1) a relearned model has similar performance to the base model, and (2) efficiently finding the relearning path relies on prior knowledge of the unlearning algorithm and forget data.

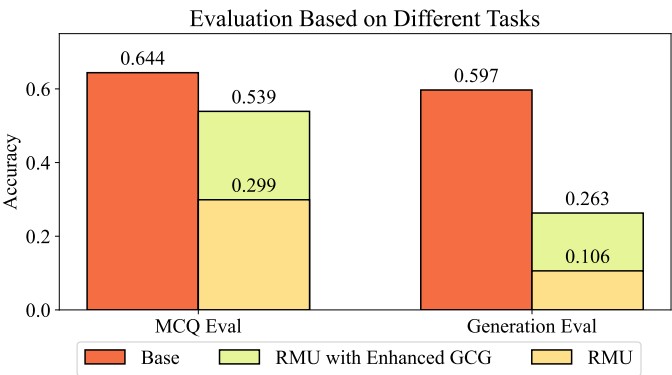

Figure 5: **Enhanced GCG task dependence.** The effectiveness of Enhanced GCG varies widely across downstream tasks. Whereas this method nearly recovers the base model's accuracy for maximum letter probability generation, it offers little improvement in accuracy for maximum text generation tasks.

on finetuning attacks implicitly assume that the data formats match between training and evaluation (see, e.g., Che et al. (2024); Maini et al. (2024); Ma et al. (2024)).

## 4.2 Input-space attacks

While it may be intuitive that finetuning evaluations inject new knowledge into an unlearned model, what may be less clear is that prompting presents another window for the inadvertent information injection. To this end, we next offer evidence that Enhanced GCG, a standard input-space attack, tends to encode information about the forget set, rendering the results of these evaluations inconclusive.

**Evidence of information injection.** As described in §3.1, Enhanced GCG optimizes a string of tokens that, when prepended to an input prompt, facilitates the elicitation of knowledge about the forget set. However, as LLMs are universal sequence approximators (Yun et al., 2019), one cannot immediately rule out the possibility of a well-optimized prefix injecting new information during evaluation. To illustrate this point, consider that in their evaluations, Łucki et al. (2024) optimize Enhanced GCG strings comprising 100 tokens on fewer than ten samples. They ultimately find that this optimization pressure is sufficient to achieve nearly 55% accuracy on the WMDP-Bio MCQ test set. This split of WMDP-Bio contains fewer than 1300 samples, and therefore encoding each of the four possible answers to each MCQ question requires approximately $1300 \times \log 4 \times 0.55 \approx 1430$ bits. On the other hand, given that the vocabulary size of the model they use is nearly $32,000$, the 100-token adversarial strings encode roughly $100 \times \log 32000 \approx 1500$ bits of information. This estimate suggests that the adversarial prompts may already embed sufficient information to elicit the correct answers, undermining the validity of the evaluation. It is thus unclear whether the prompt is reactivating superficially unlearned knowledge or introducing new knowledge into the model.

To support this hypothesis, we show that there exist prompts—specifically, Enhanced GCG prefixes—that artificially boost performance on the forget set. To this end, we first finetune Llama-3.2-1B-Instruct to the point of overfitting on the WMDP-Bio test set. We then optimize an Enhanced GCG prefix on this finetuned model. Whereas the base model achieves 39.9% accuracy on the entirety of WMDP-Bio, we find that when the optimized prefix is added to the same queries, the accuracy of the same base model reaches 53.8%. This jump suggests that there exist prompts that contain sufficient information to teach an unlearned the correct answers.

### 4.3 A common confounder: MCQ formatting

Many unlearning evaluations use data formatted as MCQs (Li et al., 2024b). While compact and intuitive, we find that *both* finetuning and input prompt evaluations can lead to inconclusive results depending on how the answers to each MCQ are calculated. To demonstrate this, assume that we are given a model $M$, an MCQ string $q$, and four answer strings $a_1, a_2, a_3$, and $a_4$. Define

$$p = \{q\}\backslash\texttt{nA.}\{a_1\}\backslash\texttt{nB.}\{a_2\}\backslash\texttt{n C.}\{a_3\}\backslash\texttt{nD.}\{a_4\}\backslash\texttt{nAnswer:} \tag{3}$$

For example, an MCQ from WMDP-Bio formatted as in the same wasy as $p$ is as follows:

> The following are multiple choice questions (with answers) about biology. Which animal model was used in the study to assess chronic renal insufficiency?\nA. Pigs\nB. Goats\nC. Rats\nD. Mice\nAnswer:

There are (at least) two reasonable ways to calculate the model's answer to $q$. The first way involves the calculation of what we term the *maximum letter probability*, which is computed by solving

$$\arg\max_\ell \ \Pr(\ell|p; M) \quad \text{subject to} \quad \ell \in \{\texttt{A}, \texttt{B}, \texttt{C}, \texttt{D}\}, \tag{4}$$

where $\Pr(\ell|p; M)$ denotes the probability of the letter $\ell$ being the next token predicted by $M$ in response to the input prompt $p$. In this way, equation 4 represents the model's greedy prediction of the letter corresponding to the correct answer. A second reasonable way to compute the answer is to calculate the *maximum text probability*, which is computed by solving

$$\arg\max_a 1/|a| \cdot \log \Pr(a|q; M) \quad \text{subject to} \quad a \in \{a_1, a_2, a_3, a_4\}. \tag{5}$$

**Inconclusive finetuning attacks.** In the context of finetuning attacks, the results in Figure 6 (left panel) show that unlearned models evaluated by computed the maximum letter probability tend to recover the base model's MCQ accuracy on WMDP-Bio. However, when answers to the same set of questions are computed via the maximum text probability, Figure 6 (right panel) shows an analogous trend does *not* hold, particularly for NPO models. This implies that MCQ evaluation accuracy is highly dependent on the method used to calculate the answers.

**Inconclusive input-space attacks.** The results in Figure 5 show an analogous trend for input prompt attacks. When answers are calculated via the maximum letter probability, we find that Enhanced GCG evaluations record 53.9% accuracy on the WMDP-Bio test set. However, when the answers are computed by validating the open-ended generation of the model, the accuracy after applying the Enhanced GCG prefix only results in 26.3% accuracy on the same dataset. This suggests that Enhanced GCG yields prefix strings, or input-space attacks, that are sensitive to the output format. Thus, we cannot conclude that some information is really present in the model, as these Enhanced GCG attacks may be overstating knowledge content by boosting MCQ accuracy only.

**Inconclusive memorization detectors.** We next design a set of analogous experiments for the ACR memorization detector described in §3.1. Specifically, we consider three Zephyr-7B checkpoints: the base model and two unlearned models obtained by running RMU and NPO. We evaluated these models on the first 100 questions from WMDP-Bio, WMDP-Chem, and WMDP-Cyber. Answers to the WMDP MCQ questions were generated in three ways: (1) the maximum letter probability as described in equation 4 (termed "CHOOSE"), (2) a variant of the maximum letter probability where we determine whether the correct answer has the highest log-probability over the model's entire vocabulary (termed "OPTION"), and (3) determining whether the response generated via greedy decoding corresponds to the text of the correct answer (termed "GENERATE"). For each combination of model, dataset, and downstream task, we use the ACR metric to compute a minimal length suffix for each sample that maximizes the probability of generating the correct answer. The evaluation is considered successful if the length of the suffix is less than a fixed task-specific threshold.

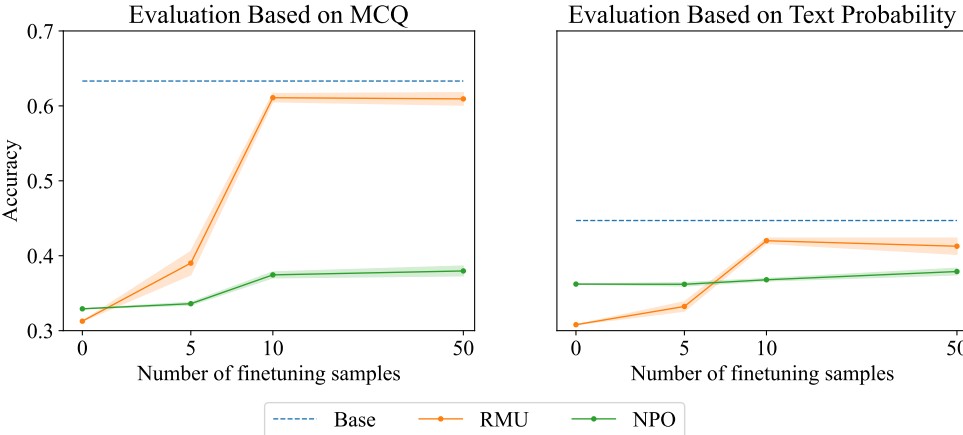

Figure 6: **Finetuning attacks across MCQ formats.** While finetuning attacks computed for maximum letter probability generation recover the base model's performance, analogous attacks on the maximum text probability do not yield similar accuracy recovery.

The results of this experiment, which are shown in Figure 7, indicate that conclusions regarding the relative effectiveness of unlearning algorithms can be drastically different across downstream tasks. For the CHOOSE task, both methods offer a relatively small, though non-negligble reduction in the success rate. In contrast, for the OPTION task, both RMU and NPO significantly reduce the success rates. And finally, the results for the GENERATE task indicate that RMU decreases the success rate, whereas NPO increase the success rate relative to the base model. Collectively, these results show that conclusions regarding unlearning vary widely depending on the downstream task.

## 5 Principles for conclusive unlearning evaluations

The evidence presented in §4 indicates that existing unlearning evaluations are often inconclusive. To support the development of more effective evaluations in future research, we propose two guiding principles—*minimal information injection* and *downstream task awareness*—which we summarize below and describe in more detail in the ensuing section.

> **Principles for conclusive unlearning evaluations**
>
> 1. **Minimal information injection**: Unlearning evaluations should not enable the injection of additional information into an unlearned model.
> 2. **Downstream task awareness**: Unlearning evaluations should be designed to accommodate the ways in which future users will interact with an unlearned model.

**Minimal information injection.** In §4, we observed that both finetuning attacks and input-prompt attacks directly inject new information into an unlearned model. Thus, we argue that evaluations should minimize the amount of information unintentionally injected into the model. This principle—which we term *minimal information injection*—ensures that the generations of an unlearned model are a realistic reflection of the model's knowledge, rather than an artifact of the evaluation process.

**Downstream task awareness.** In §4, we also found that standard unlearning evaluations yield varying results depending on the task in which the unlearned model may ultimately be deployed. More specifically, we found that unlearning results varied depending on whether a model was evaluated via open-ended generation, maximum letter probability, or maximum text probability. We therefore advocate that unlearning evaluations be designed to accommodate the various ways in which future users will interact with an unlearned model, a property we term *downstream task awareness*.

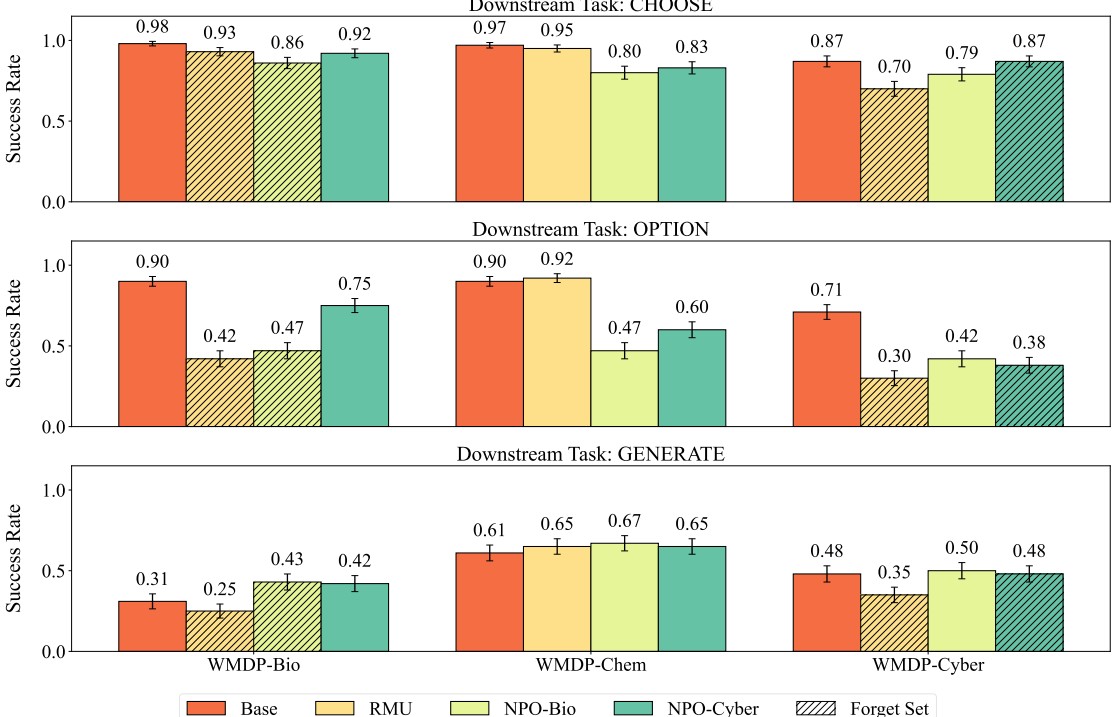

Figure 7: **Inconclusive ACR evaluations on WMDP-Bio.** Success rates of different models on different datasets and tasks. The error bars show the standard errors. For each task, the conclusion about the relative effectiveness of unlearning methods differ. **Top:** CHOOSE task. The success rates of the base and unlearned models are relatively similar; unlearning tends to decrease the success rate marginally. **Middle:** OPTION task. The success rates of the unlearned models are significantly lower than the base model. **Bottom:** GENERATE task. The RMU model has a lower success rate than the base model on WMDP-Bio and WMDP-Cyber. However, NPO tends to increase success rates.

### 5.1 Recommendations for future evaluations

**Recommendation 1: Disclosure of an "injection budget."** Throughout this work, we measure information injection by counting the number of bits available to an attacker and by measuring spurious generalization during relearning. Both of these metrics are relatively heuristic, which is indicative of the inherent difficulty in accurately quantifying how efficiently an LLM processes and stores information (Gekhman et al., 2025). We therefore argue that future work should seek to measure injected information, such as the heuristic measures or other tools in the information theoretic literature surrounding LLMs (Pimentel et al., 2020; Chen et al., 2024; Pezeshkpour, 2023).

While we frame this recommendation in general terms, the selection of an injection budget is often specific to the unlearning task and corresponding evaluation metrics. Therefore, to illustrate how this recommendation can be operationalized in practice, we walk through two example scenarios.

- **Example scenario 1: Finetuning attacks for MCQ datasets.** In §4.1, we argued that finetuning on retain set data can spuriously generalize (e.g., due to stylistic patterns) to improve performance on the forget set. In this scenario, we recommend instantiating Recommendation 1 via the disclosure of a *relearning generalization index*, which we define as

$$\text{RGI}(\text{Accuracy}_{\text{forget}}, N) = 100 \times \frac{\Delta \text{Accuracy}_{\text{forget}}}{N},$$

where $\Delta\text{Accuracy}_{\text{forget}}$ is the increase in forget set accuracy after finetuning relative to the unlearned baseline, and $N$ is the number of finetuning samples. For instance, if the unlearned model has 30% accuracy on the forget set, and finetuning on 100 retain examples increase this to 50%, then $\text{RGI} = 0.2$, meaning that each example contributes 0.2 percentage points of forget set accuracy on average. We therefore recommend capping $N$ and disclosing the RGI. We also recommend ensuring that $\text{RGI} \geq \tau$ for a problem dependent threshold $\tau \geq 0$ (in our example, $\tau = 0.2$ is reasonable) to ensure that finetuning evaluations do not rely on a large set of examples which inject significant amounts of information into the unlearned model.

- **Example scenario 2: Input-space attacks for MCQ datasets.** In §4.2, we argued that a 100-token Enhanced GCG string can carry up to ∼1500 bits of information, potentially enough to encode answer keys for the entire forget set. More generally, for a MCQ dataset with $N$ questions and $K$ choices per question, encoding all answers requires $N \log K$ bits, while an adversarial prompt of length $L$ with tokens from a vocabulary of size $V$ can encode up to $L \log V$ bits. In this scenario, we recommend instantiating Recommendation 1 via the disclosure of a *prompt-data entropy ratio* (PDER), which we define as follows:
$$\text{PDER}(N, K, L, V) = \frac{L \log V}{N \log K}.$$

  Ideally, $\text{PDER} \ll 1$, indicating an input prompt cannot encode all answers. For instance, in WMDP-Bio, $N \approx 1300$ and $K = 4$, so encoding all answers requires about 2600 bits. If the adversarial prompt has length $L = 100$ and the vocabulary size is $V \approx 32000$, then the prompt can encode up to about 1500 bits, yielding $\text{PDER} \approx 0.58$. This value is too high, as it indicates that the prompt may be able to encode a large fraction of the answers. In the ideal case, $L$ should be chosen to be a smaller value, such as $L = 10$, yielding $\text{PDER} \approx 0.06 \ll 1$.

**Recommendation 2: Report cross-modality metrics.** Task sensitivity in unlearning evaluations is indicative of a wider trend; recent findings have noted similar robustness concerns in tasks beyond unlearning (Sclar et al., 2023; Zheng et al., 2023; He et al., 2024). Indeed, task sensitivity is not necessarily a pitfall in tasks where a particular data format is generally preferred. However, the goal of unlearning is to ensure that the model cannot generate any information from the forget set, regardless of its format. Therefore, in line with the downstream task awareness principle, we recommend that future evaluations report cross-format metrics, which, as in Figure 7, probes an unlearned model for unlearned information across data formats. This standard contributes to a research environment in which describing a model as "unlearned" is more compelling, because a model that can still generate forgotten information in a different format has not truly unlearned.

## 6 Conclusion

In this work, we identify two shortcomings in existing adversarial evaluations—unintended information injection and dependence on data format—that undermine their reliability. To mitigate this, we propose two guiding principles for more reliable assessment methods. We encourage future research to develop more efficient and principled evaluation techniques.

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

# A    Additional Experiments

In this section, we present additional results in the set of experiments described in Section 5. To interpret the results, we note that the unlearned models we use in Section 5 are only unlearned on a subset of the WMDP dataset. Specifically, the RMU model is unlearned on both WMDP-Bio and WMDP-Cyber, while the NPO model is unlearned on WMDP-Bio. In the full experiments, we include another model that is unlearned on WMDP-Cyber using the NPO method. We differentiate the NPO models unlearned on WMDP-Bio and WMDP-Cyber by referring to them as NPO-Bio and NPO-Cyber, respectively.

## A.1    Success Rates for Different Downstream Tasks

In Fig. 7, we show the success rates on WMDP for 3 different downstream tasks mentioned in Section 4.3. The results confirm that conclusions about the effectiveness of unlearning methods can be drastically different for different downstream tasks. For the CHOOSE task, both RMU and NPO reduce the success rate on the datasets they unlearn on. However, neither of them decrease it significantly. For the OPTION task, both RMU and NPO reduce the success rate significantly, with RMU reducing it slightly more. For the GENERATE task, RMU reduces the success rate, while NPO increases it.

On a side note, we also remark that none of the models are unlearned on WMDP-Chem, so we could consider WMDP-Chem as a retain set. As we can see, for the CHOOSE and OPTION tasks, the success rate of the RMU model on WMDP-Chem is similar to the base model, while the NPO models have lower success rates on WMDP-Chem. This suggests that RMU preserves the model's performance on benign knowledge better than NPO.

## A.2    Adversarial Compression Ratio for Different Downstream Tasks

In Table 1, we show the 40%, 50% and 60% percentiles of adversarial compression ratio (ACR) of the models on WMDP for the GENERATE task. The results show that RMU has a better effect of unlearning. The RMU model has a lower ACR than the Base model on WMDP-Bio and WMDP-Cyber. The NPO models also reduce ACRs, but the magnitude is smaller. This indicates that it is harder to extract hazardous knowledge from the RMU model, which concurs with the results in Fig. 7. Moreover, looking at the results of WMDP-Chem–data that is not included in any forget sets in this experiment–we can also see that the RMU model and the Base model's performance are consistently relatively similar on knowledge that is not targeted in the unlearning process, while the NPO models tend to decline in performance on such knowledge. This suggests that RMU more effectively preserves the model's performance on benign knowledge. Overall, when considering open-ended generation as the downstream task, RMU does seem to have a better effect of unlearning.

Table 1: 40%, 50% and 60% percentiles of adversarial compression ratio (ACR) of 4 variants of Zephyr 7B on WMDP for the GENERATE task. Performance is compared against Base. Green cells highlight the models unlearned on the corresponding datasets. Higher intensity of the color means it is harder to extract the knowledge. The results show that RMU has a better effect of unlearning.

| Dataset | Base | RMU | NPO-Bio | NPO-Cyber |
|---|---|---|---|---|
| WMDP-Bio | 2.00 / 2.50 / 2.87 | 0.85 / 1.25 / 1.87 | 1.50 / 1.58 / 1.87 | 1.50 / 2.00 / 2.00 |
| WMDP-Chem | 1.50 / 1.67 / 2.00 | 1.50 / 1.67 / 1.77 | 1.50 / 1.57 / 1.67 | 1.43 / 1.63 / 2.00 |
| WMDP-Cyber | 2.38 / 3.00 / 3.00 | 1.56 / 1.83 / 3.00 | 2.00 / 2.10 / 2.70 | 2.20 / 2.75 / 3.00 |

# B    Experiment Details

**Hyperparameters.**    For the experiments involving finetuning on WMDP data in Section 3.1, we always use LoRA rank 128, $\alpha = 16$, learning rate 2e−4, and we always train for 3 epochs with batch size 1. For the experiments with ACR in Section 3.1, we use learning rate 1e−2, batch size 100, and the 250 top choices for the optimization step. For the CHOOSE and OPTION tasks, we run 200 steps of optimization with 5 free

tokens. For the GENERATE task, we run 350 steps with 20 free tokens. We choose these thresholds such that further increasing them does not boost the success probability of the corresponding task significantly further. For the TOFU experiment, we train with LoRA rank 128, $\alpha = 16$, learning rate 2e−4, and we always train for 5 epochs with batch size 16.

**Computation.** For all our experiments, we use one A6000 with 48GB of memory. The experiments in Section 4 take around 20 GPU hours, and those in Section 5 take approximately 800 GPU hours.

**Data curation.** To curate TOFU-MCQ, we first took the (original) TOFU-QA dataset and converted it to a Wikipedia format article by prompting ChatGPT. In particular, TOFU-QA contains 200 authors, each has 20 QA pairs. For each author, we generated one Wikipedia-like article. Then we asked ChatGPT to generate 10 MCQs for each article. We shuffled the options for each MCQ to ensure that the distribution of the correct answer was nearly uniform.

