# OpenReview forum: "Existing Adversarial Large Language Model Unlearning Evaluations Are Inconclusive"
_TMLR — Withdrawn by Authors_

### Review · Reviewer_pQmg · 2026-02-15

**Summary Of Contributions:**

The paper studies the current adversarial based evalution on the machine unlearning tasks. Specifically, it studies fine-tuning attacks, Input-space attacks and memorization detectors. The study conducted on several models and unlearning methods shows that both the fine-tuning attacks and input-space attacks will introduce new information about the forget set and the format matters a lot, which makes the evaluation not reliable. The ACR memorization detectors are also very sensitive to the format. The paper proposes two recommendations on the unlearning evaluation: two metrics and with diverse downstream tasks.

**Audience:**

Yes

**Audience Explanation:**

The machine unlearning is an important research question and the evaluation of the unlearning performance is essential to compare different methods' effectiveness. The findings on incorrect evaluation with adversarial methods give the community a way to rethink the problem and can introduce a better evaluation in the future.

The main concern I have is that the recommendation is not concrete enough for the practitioner to use and it is relatively a broad guideline. The two metrics proposed are also not tested in their own evaluation pipeline and, thus, are not convincing enough.

**Broader Impact Concerns:**

No ethical concerns for me.

**Claims And Evidence:**

Yes

**Claims Explanation:**

The issues with the adversarial evaluations have been studied with a series of controlled studies. The findings are interesting and well-established.
One concern I have is that the methods have only been tested in two unlearning methods, and only the MCQ format has been studied. Although some evidence supports the hypothesis in the paper, the claim could only be applicable to the two studied unlearning methods.
I would like to see some discussion with non adversarial evaluation method. If the format also matters a lot in these evaluations, it is difficult to make the paper's claim stand out.

**Requested Changes:**

1. Add more unlearning methods to strengthen the paper's claim.
2. Have some discussion with a non-adversarial evaluation in the studied setting.
3. Make the recommendation more concrete and test their proposed metrics with their pipelines.

---

### Review · Reviewer_sond · 2026-02-21

**Summary Of Contributions:**

This paper aims to analyze the efficacy of existing evaluations of LLM unlearning. The analysis focuses on adversarial evaluations, namely:
* fine-tuning attacks (can an attacker choose fine-tuning samples such that "forbidden" info is learned)
* input space attacks (can "forbidden" info be elicited via prompting)
* memorization attacks (adversarial compression ratio; i.e, how much "forbidden" info can be extracted per "unit" of prompting "effort?")

The work claims that there are two barriers impeding evaluations of LLM unlearning: (1) the evaluation strategy itself may reveal information to the model, and (2) evaluation metrics for measuring unlearning may not capture all possible use cases. They validate these claims as follows:
* Claim 1:
    * 1.1. — For fine-tuning attacks, Phi-1.5 is found to generalize on TOFU, a popular unlearning benchmark, suggesting spurious correlations that aid generalization on the forget set.
    * 1.2 — Re: FT attacks —relearning is faster when the forget set data format matches that of the retain set data.
     * 1.3 — Input-space attacks are confounded by the fact that it's ambiguous whether an injected prefix is responsible for eliciting "forget set" knowledge, or whether the prefix itself has induced new knowledge.
* Claim 2:
    * 2.1 — choice of evaluation metric (the example used in the paper is MCQ w/ maximum letter probability vs. maximum text probability) can also confound conclusions re: unlearning.

**Audience:**

Yes

**Audience Explanation:**

The papers hypotheses approximate a well-scoped empirical study of LLM unlearning. While many of the findings of this paper may not be surprising, this is not a reason to reject if the scope differs from previous empirical studies. I find pieces that offer methodological guidance rather than proposing new methods under-appreciated at times and would advocate for inclusion of such works.

**Claims And Evidence:**

No

**Claims Explanation:**

This is a well-targeted empirical study of LLM unlearning. I find the evidence mostly clear and accurate as far as I can tell, but not as convincing due to lack of breadth — please see my suggestions below.

**Requested Changes:**

Ultimately, while I like the aims of the paper, I think there needs to be significant improvement in the experimental design & execution of the work before I advocate for acceptance.

**Critical**
* My main issue is that, while the hypotheses in this paper are well formed, they're tested on a limited subset (often one) of models and datasets. While no paper can test every permutation of models/datasets (+ too many models/datasets w/ no reason is also not good), it might be useful to set expectations upfront about which models and datasets are likely to yield insights about the underlying task.
    * As-is, I have to believe that all models/unlearning datasets are essentially exchangeable for the results to hold, which is a big stretch. Are there classes of models or styles of tasks that the authors would consider adding? Why would they be informative in a study about unlearning?
* Furthermore, the proposed guidelines for new unlearning evaluation are nice — do they hold up on the experiments shown in the paper? What happens if we re-evaluate unlearning using the proposed framework?
* Can you provide practical examples of how one would implement Recommendation 2 (Report cross-modality metrics), similar to those provided in Recommendation 1?
    * Nit: cross-modality makes me think of something like unlearning in text space -> elicitation in image space — is "cross-format" or "multi-format" the intended meaning? This is more in line with Section 4.3.
* I'm very confused about the claim re: data formatting dependence — it seems to go something like "relearning + eval on A yields higher performance than relearning on A + eval on B," which seems like a straightforward result? It's unclear to me how this result gives us insight into the limitations of unlearning evaluations.
    * If the claim is "current unlearning evaluations assume relearning format == eval format; **perhaps we shouldn't assume this**" — the last part of that assertion isn't made clear.

**Nice to have**
* Re: Recommendation 1 — any guidance on what budgetary "units" are suitable? Bits are mentioned, but one can also consider # of tokens, wall-clock time (e.g., time to search for a prompt injection), or others.
* Re: Recommendation 2 — this recommendation seems somewhat ill-scoped, since aren't there an infinite number of modalities/formats where knowledge could be found? Would be nice to see this limitation mentioned.

Also, a major suggestion — I wonder if this paper would benefit from a larger restructure. Because the empirical study on unlearning is the first result, I'm primed into expecting a more comprehensive probe of how different models/data respond under the various unlearning evaluation methods mentioned in the intro. However, there's a case of leading with something like the curr. Section 5, making proposal of a framework for unlearning evaluation the main contribution (for which Section 4.3 is a really *great* practical example), and then the rest of the work is a validation that the proposed framework provides "better" insights about unlearning than existing evaluations.

To that end, rather than showing that unlearning methods can fail to achieve unlearning in practice, far more interesting to me would be if the authors can show that problems with current unlearning evaluations could actually lead us to contradictory conclusions about unlearning efficacy.

---

### Review · Reviewer_YjAQ · 2026-03-03

**Summary Of Contributions:**

This paper critically examines adversarial evaluation methods for large language model unlearning and argues that current evaluation practices yield inconclusive results. Specifically, the authors analyze three widely used classes of adversarial evaluations: finetuning attacks, input-space attacks (e.g., Enhanced GCG), and memorization detectors (e.g., adversarial compression ratio), and identify two core shortcomings:

1. Information injection during evaluation: The evaluation procedure itself may introduce new information into the model, for example, via finetuning on correlated retain data or via high-capacity adversarial prompts, thereby obscuring whether apparent knowledge reflects failure to unlearn or re-teaching during evaluation.
2. Task-format sensitivity: Evaluation outcomes vary substantially across downstream task formats, such as multiple-choice letter prediction, maximum text probability, and open-ended generation, leading to contradictory conclusions about unlearning effectiveness.

Through controlled experiments on TOFU, WMDP, and multiple model architectures, including Zephyr-7B, Phi-1.5, and Llama-3.2-1B-Instruct, the paper shows:

* Spurious generalization from retain to forget sets in finetuning attacks.
* Capacity of adversarial prefixes such as 100-token Enhanced GCG prompts to encode sufficient information to answer MCQs.
* Large discrepancies in unlearning conclusions depending on the computation method.
* Divergent conclusions from adversarial compression ratio-based memorization detection depending on downstream task formulation.

Based on these findings, the authors propose two guiding principles for future evaluations:

* Minimal Information Injection
* Downstream Task Awareness

They further operationalize these principles through concrete metrics such as the Relearning Generalization Index (RGI) and the Prompt-Data Entropy Ratio (PDER), and recommend cross-format reporting.

Key strengths:

* Timely and important meta-evaluation of unlearning research.
* Clear identification of conceptual flaws in widely used benchmarks.
* Strong empirical evidence across multiple models and datasets.
* Concrete, actionable recommendations rather than purely conceptual critique.
* Thoughtful information-theoretic framing of prompt capacity.

Key weaknesses:

* The proposed principles are high-level and not yet validated through adoption in a new benchmark.
* Some metrics such as RGI and PDER are heuristic and may not fully capture information flow.
* The paper critiques existing benchmarks but does not propose a full alternative evaluation pipeline.
* The analysis focuses primarily on MCQ-style datasets such as TOFU and WMDP, which may limit generality.

**Additional Comments:**

This is a well-executed and timely meta-evaluation of a rapidly evolving area. The paper makes a strong case that many current unlearning evaluations conflate forgetting with re-teaching or task-format artifacts. The proposed principles are sensible and likely to influence future benchmarking standards.

With modest clarifications and a stronger discussion of generality, I believe this work would make a valuable contribution to TMLR.

**Audience:**

Yes

**Audience Explanation:**

Unlearning is a rapidly growing research area due to regulatory pressure, privacy concerns, and AI safety considerations. As a result, evaluation methodology is central to progress in this domain.

This paper addresses a foundational methodological issue: whether current evaluation protocols meaningfully measure forgetting. The findings are highly relevant to:

* Researchers working on machine unlearning
* LLM safety and alignment researchers
* Benchmark designers
* Researchers concerned with memorization and data extraction
* Policymakers and practitioners interested in verifiable data deletion

Because the paper challenges prevailing evaluation norms and proposes principled alternatives, it is likely to shape future work in this area.

**Broader Impact Concerns:**

The paper discusses unlearning in the context of safety and regulatory compliance. The broader impacts are largely positive, as improved evaluation methods could:

* Prevent overclaiming of unlearning success.
* Improve compliance with privacy regulations.
* Reduce false assurances about the removal of sensitive knowledge.

However, there is a potential dual-use consideration: improved evaluation of adversarial elicitation could also inform the development of stronger attacks. That said, the paper’s main contribution is critique and methodological guidance rather than attack construction, and the benefits likely outweigh the risks.

**Claims And Evidence:**

Yes

**Claims Explanation:**

The authors provide extensive empirical evidence supporting their central claims. In Section 4, they demonstrate:

* Spurious generalization (Fig. 2): Finetuning on retain-set MCQs significantly improves held-out forget-set performance, suggesting dataset correlations undermine the independence assumption.
* Format dependence in finetuning (Fig. 3 and Fig. 6): Accuracy recovery differs dramatically depending on whether the evaluation uses maximum letter probability or maximum text probability.
* Information capacity of prompts (Section 4.2): A quantitative bit-capacity argument shows that 100-token adversarial prefixes can encode enough information to answer a large fraction of forget-set MCQs.
* Task-dependent memorization results (Fig. 7): Success rates vary widely across CHOOSE, OPTION, and GENERATE tasks, even reversing relative method rankings.

These results consistently demonstrate that conclusions about the effectiveness of unlearning depend heavily on evaluation design choices.

The empirical setup spans multiple models, unlearning algorithms such as RMU and NPO, and datasets including TOFU and WMDP, lending robustness to the claims. The paper does not rely on anecdotal evidence but instead presents systematic comparisons that directly support its arguments.

**Requested Changes:**

Critical for Acceptance:

1. Clarify Scope of Generality: The empirical analysis is largely centered on MCQ-style benchmarks such as TOFU-MCQ and WMDP. The authors should clarify whether the identified pitfalls extend to open-ended corpora-based benchmarks beyond those tested. Even a brief additional experiment on non-MCQ data would strengthen the generality claim.

2. Stronger Theoretical Framing of Information Injection: While the bit-capacity arguments are intuitive, they are heuristic. The authors should more explicitly discuss the limitations of this approximation and clarify what assumptions underlie the PDER metric.

3. Explicit Guidance on Threshold Selection: The proposed RGI and PDER require threshold choices such as τ for RGI. The paper should provide more guidance on how to select such thresholds in practice.

Non-Critical (Would Strengthen the Paper):

4. Provide a summarized checklist for evaluation design aligned with the two principles.
5. Discuss implications for non-adversarial evaluation methods.
6. Clarify computational costs relative to prior evaluation pipelines.

---

### Note · Authors · 2026-03-13

**Comment:**

We would like to sincerely thank the Action Editor and all the reviewers for their time, feedback, and constructive suggestions. We are glad to see that the reviewers found our critique of current adversarial unlearning evaluations to be timely and our empirical findings across models to be strong.

We completely agree with the reviewers' assessments that expanding the empirical scope (specifically, incorporating non-MCQ datasets) and thoroughly validating the proposed RGI and PDER metrics within our own pipeline would significantly strengthen the paper and its generality. However, doing justice to these "Critical for Acceptance" requests would require a substantial structural overhaul and a large suite of new experiments. Given the current situation, we are unable to complete these major revisions within the rebuttal time frame. Therefore, we have decided to withdraw this submission.

We deeply appreciate the rigorous insights provided by the reviewing team.

**Withdrawal Confirmation:**

I have read and agree with the venue's withdrawal policy on behalf of myself and my co-authors.